# Comparison of the Predictive Accuracy of Intraocular Lens Power Calculations after Phototherapeutic Keratectomy in Granular Corneal Dystrophy Type 2

**DOI:** 10.3390/jcm12020584

**Published:** 2023-01-11

**Authors:** Sook Hyun Yoon, Woo Kyung Jo, Tae-im Kim, Kyoung Yul Seo, Jinseok Choi, Ikhyun Jun, Eung Kweon Kim

**Affiliations:** 1Department of Ophthalmology, School of Medicine, Daegu Catholic University, Daegu 42472, Republic of Korea; 2The Institute of Vision Research, Department of Ophthalmology, College of Medicine, Yonsei University, Seoul 03722, Republic of Korea; 3Corneal Dystrophy Research Institute, Yonsei University College of Medicine, Seoul 03722, Republic of Korea; 4Saevit Eye Hospital, Goyang-si 10447, Republic of Korea

**Keywords:** granular corneal dystrophy, phototherapeutic keratectomy, IOL power calculation

## Abstract

Granular corneal dystrophy type 2 (GCD2) is an autosomal dominant disease affecting vision. Phototherapeutic keratectomy (PTK) is advantageous in removing vision-threatening corneal opacities and postponing keratoplasty; however, it potentially disturbs accurate intraocular lens (IOL) power calculation in cataract surgery. The myopic/hyperopic Haigis-L method with or without the central island has been reported; nevertheless, an optimal method has not yet been established. To compare the predictive accuracy of post-PTK IOL power calculations in GCD2, the retrospective data of 30 eyes from July 2017 to December 2020 were analyzed. All GCD2-affected eyes underwent post-PTK standard cataract surgery using the WaveLight EX500 platform (Alcon Laboratories, Inc., Fort Worth, TX, USA) under a single surgeon. The mean prediction error (MPE) and absolute error (MAE) with the myopic/hyperopic Haigis-L, Barrett Universal II, Barrett True-K, Haigis, and SRK/T by standard keratometry (K) and total keratometry (TK), where possible, were analyzed. Barrett Universal II and SRK/T showed significantly superior MPE, and MAE compared with the myopic/hyperopic Haigis-L method. TK was not significantly superior to K in the same formula. In conclusion, this study suggests that these biometries and formulas, especially Barrett Universal II and SRK/T, are potentially useful in IOL power calculation in GCD2 after PTK.

## 1. Introduction

Granular corneal dystrophy type 2 (GCD2) is a common autosomal dominant corneal stromal dystrophy in Korea, with an estimated prevalence rate of 1/870 [1]. GCD2 exhibits three types of corneal deposits: granular deposits, lattice deposits, and diffuse stromal haze with progression in heterozygotes [2]. Visual acuity is maintained relatively well as the deposits remain visual-axis clear [3]. With age, however, diffuse stromal haze develops around the central cornea, and phototherapeutic keratectomy (PTK) is a favorable option for removing vision-threatening corneal opacities and postponing keratoplasty [4].

Despite its necessity to improve vision, PTK potentially disturbs accurate intraocular lens (IOL) power calculation in cataract surgery because of changing corneal curvature [5]. It renders the refraction index of 1.3375 inapplicable and the calculation of effective lens position inaccurate. Yaguchi et al. suggested the Camellin–Calossi and PhacoOptics formulas as favorable options [6,7]. However, Yoneyama et al. reported that the total corneal refractive power provided the highest predictability in post-PTK eyes [8]. Nonetheless, these reports did not focus on patients with GCD2.

In previous data, the myopic/hyperopic Haigis-L formula with topographic analysis for central island formation was a plausible method considering the underestimation associated with IOLMaster^®^ keratometry [9]. Recently, the swept-source optical coherence tomography (SS-OCT) biometer (IOLMaster 700, Carl Zeiss Meditec AG) was found to provide total keratometry (TK) and more accurate IOL power calculation in patients who underwent refractive surgery [10,11]. An additional benefit may be expected from newly developed formulas, such as the Barrett True-K formula. Furthermore, a newly developed laser system, namely, the WaveLight EX500 platform (Alcon Laboratories, Inc., Fort Worth, TX, USA), potentially affects post-PTK keratometry (K).

Therefore, this study aimed to compare the postoperative refractive errors resulting from these formulas after cataract surgery in post-PTK eyes with GCD2 using standard K and TK.

## 2. Materials and Methods

### 2.1. Patients

The protocol of this retrospective study was approved by the Institutional Review Board of the Yonsei University College of Medicine (IRB no. 4-2022-0292) and was conducted in accordance with the tenets of the Declaration of Helsinki. Thirty eyes of 28 patients with heterozygous GCD2 who had undergone a post-PTK cataract surgery at Severance Eye hospital in Seoul, Korea, between July 2017 and December 2020 were included. The inclusion criteria were eyes with in-the-bag IOL implantation cataract surgery after PTK. GCD2 was confirmed in all patients by DNA analysis, and they underwent cataract surgery at least 4 months after PTK.

The exclusion criteria were as follows: patients with any medical history affecting corneal curvature, including ocular surface disease or contact lens wear; any previous procedures using an excimer laser, except a single PTK procedure performed before cataract surgery; and complications during cataract surgery, such as posterior capsule rupture.

### 2.2. Surgical Procedures

All PTKs were performed by one surgeon (E.K.K) using the WaveLightEX500 (Alcon Laboratories, Fort Worth, TX, USA), with a radiant exposure of 160 mJ/cm^2^. The ablation diameter was 6.0 mm without a transition zone. The density and depth of the remaining diffuse stromal haze were verified using a slit lamp after each 10–20-µm ablation to prevent overtreatment. For ablation smoothness, preservative-free carboxymethylcellulose sodium 0.5% solution (Refresh Plus; Allergan Inc, Irvine, CA, USA) was instilled at a rate of one drop after every 10-µm ablation. PTK was performed until the diffuse stromal haze disappeared, and the total ablation depth was recorded. At the end of ablation, additional photorefractive keratectomy (PRK) was performed. After ablation, 100% serum eyedrops were applied four times per day until epithelial healing was complete. Topical ofloxacin 0.3% (Ocuflox; Samil Pharmaceutical Co., Seoul, Korea) and fluorometholone 0.1% (Ocumethelone; Samil Pharmaceutical Co.) were applied four times per day for 1 month and tapered within 4 months.

Additional PRK after PTK was performed owing to the ablation difference between the central and peripheral zones by the ablation profile of WaveLightEX500. The ablation difference caused by a deeper peripheral zone and the shallow central zone was neutralized by an equal amount of additional myopic PRK (Appendix A).

Cataract surgery was performed by the same experienced surgeon at least 4 months later using standard phacoemulsification with a 2.8-mm temporal corneal incision after pin-point anesthesia. In-the-bag IOL implantation was performed using the Tecnis ZCB00 IOL (Johnson & Johnson Vision, Inc., Santa Ana, CA, USA).

### 2.3. Topographic Analysis for Central-Island Estimation

Corneal topography before cataract surgery was analyzed using the Pentacam^®^ (Oculus, Wetzlar, Germany) to detect central islands after PTK. According to a previous study, we considered that the central island was present if the difference was ≥3.00 D between the central power and the mean power of six points proximal to the 4.0-mm zone [9]. Krueger et al. suggested the central island formation as an area with a diameter > 1.5 mm, having high refractive power when compared with the surrounding area of reduced topographic power [12].

### 2.4. IOL Power Calculation Methods

All patients underwent a full ophthalmic examination before cataract surgery, including slit-lamp examination, intraocular pressure measurement, indirect ophthalmoscope examination, manifest refraction, and autokeratometry (KR-8800, Topcon, Tokyo, Japan). Axial length, anterior chamber depth, central corneal thickness, white-to-white, lens thickness, K, and TK were measured using SS-OCT.

The IOL power was calculated based on preoperative data using the following formulas: the myopic Haigis-L method if there was no central-island representation, the hyperopic Haigis-L method if a central island was present [9], and Haigis, Barrett Universal II Barrett True-K, and SRK/T. K and TK were applied to all formulas using the IOL Master 700, excluding the Barrett True-K formula.

Two keratometries were employed under SS-OCT. K was calculated based on the anterior corneal curvature from measuring reflections of 18 light-emitting diodes combined with telecentric keratometry. In contrast, TK was calculated based on the anterior corneal curvature, posterior corneal curvature, and corneal thickness derived from the combination of telecentric keratometry and optical coherence tomography technology.

### 2.5. Evaluation of the Predictive Abilities of the IOL-Power-Calculation Methods

The evaluation was performed using the mean prediction error; absolute error; and proportion of eyes within ±0.25 D, ±0.50 D, and ±1.00 D of the predicted postoperative spherical equivalent refraction at 2 months after surgery. The mean absolute error was defined as the absolute value of the prediction error minus the postoperative spherical equivalent of the manifested refraction from the target IOL diopter values.

### 2.6. Statistical Analysis

Statistical analyses were performed using SPSS statistical software (version 25; IBM^®^, Armonk, NY, USA). The Kruskal–Wallis H test was used to analyze the prediction and absolute errors using several IOL-power-calculation methods together with Bonferroni post-hoc analysis. The results are expressed as means ± standard deviations, and probability values < 0.05 were considered significant.

## 3. Results

In total, 30 eyes of 28 patients with GCD2 after PTK were included in this study. The mean age at cataract surgery was 68.50 ± 6.84 (57–78) years, and the mean interval between PTK and cataract surgery was 8.37 ± 7.55 (4–33) months. The mean IOL power for cataract surgery was 22.37 ± 3.23 (15–28) D. Patient characteristics and preoperative biometry parameters according to the IOLMaster is presented in Table 1.

Table 2 shows the mean prediction and absolute errors of the predicted postoperative spherical equivalent refraction at 2 months after surgery. Two eyes had central islands, according to topographic analysis. Therefore, the IOL power calculations of these two eyes used the hyperopic Haigis-L formula, whereas those of the other 28 eyes without central islands used the myopic Haigis-L formula.

The prediction errors according to the Barrett Universal II and SRK/T K formulas were significantly smaller (*p* = 0.027, *p* = 0.045) than those according to the myopic/hyperopic Haigis-L K. No significant differences between other methods were observed (Figure 1).

The absolute error was not significantly different between the Barrett True-K and myopic/hyperopic Haigis-L K formulas. However, the Haigis K, Haigis TK, Barrett Universal II, Barrett TK Universal II, SRK/T K, and SRK/T TK formulas were significantly superior to the myopic/hyperopic Haigis-L K method (*p* = 0.002, *p* = 0.08, *p* = 0.034, *p* = 0.002, *p* = 0.002, *p* = 0.002, and *p* = 0.007, respectively). No significant differences were noted between K and TK in the same formula, excluding the myopic/hyperopic Haigis-L formula (Figure 2).

Table 3 and Figure 3 show the proportions of eyes within ±0.25 D, ±0.50 D, and ±1.00 D of the predicted postoperative spherical equivalent refraction at postoperative 2 months. All formulas except the Barret True-K formula demonstrated higher proportions of eyes within ±0.5 D than the myopic/hyperopic Haigis-L method. The Barrett Universal II and Barrett TK Universal II formulas had 60% of eyes within ±0.75 D, whereas the SRK/T K formula exhibited the highest proportion within ±1.0 D.

## 4. Discussion

Central island formation after PTK is one cause of IOL power-calculation disturbance. In a previous study, the degree of central island formation was not significantly correlated with the residual corneal thickness or ablation depth [13]. Several possible mechanisms of central island formation have been suggested. Regional differences in corneal epithelial healing, laser energy absorption by a plume of ablative products, and non-uniformity of the excimer laser beam due to degraded laser optics have been advocated [12,14,15]; nevertheless, the exact mechanisms remain unknown.

The laser ablation system has recently been proposed as the possible cause of central island formation. Using a broad-beam excimer laser system (VISX S4 IR, Santa Clara, CA, USA), central islands were detected in 14/20 eyes [9]. Accordingly, Hashimoto et al. detected central islands in 6/11 eyes (55%) with granular dystrophy after PTK using the same laser system; they proposed the necessity of an anti-central island software application for PTK in a clinical setting [13]. Recently, the flying-spot excimer laser was found to be associated with a smaller incidence of central island formation than the broad-beam excimer laser [13,16,17]. Central island formation in this study was detected in two eyes using the flying-spot laser system when the additional PRK was applied to neutralize the difference in the ablation thickness between the central and peripheral cornea.

This study demonstrated that the Barrett Universal II and SRK/T K formulas were both significantly superior to the myopic/hyperopic Haigis-L K method in terms of mean prediction and absolute errors. Yaguchi et al. reported that the SRK/T formula was associated with significant hyperopic shifts compared with the Haigis-L method [6], which was consistent with our findings. Yaguchi et al. recently suggested that PhacoOptics with the C-constant method and the Camellin–Calossi formula, which is a newer ray-tracing method, were favorable options for calculating IOL power [7]. However, in addition to studying eyes with GCD2, these two studies also focused on other ocular diseases, such as bullous keratopathy. Therefore, their results may not be fully generalizable to GCD2 alone.

In general, TK, integrated into the IOLMaster 700 [11], provides more accurate information regarding total corneal power in astigmatic or post-refractive surgery [10,18,19,20]. In this study, however, TK use was not superior to K in the same formula. The SRK/T and Barrett formulas reportedly produced more accurate results when using conventional K in cataract surgery without previous PTK [21]. TK may not always be suitable for all formulas. Hypothetically, the remnant deeper opacity in GCD2 after PTK potentially disturbs posterior curvature estimation by SS-OCT. The smooth surface, according to the advanced laser system, probably made a similar curvature to the naïve pre-PTK curvature, thus potentially rendering K a more useful alternative to TK.

To estimate the post-PTK curvature more precisely, Yoneyama et al. reported that the SRK/T formula with total corneal refractive power (TCRP) using a rotating Scheimpflug camera was superior to automated and simulated keratometry in IOL calculation in post-PTK cataract surgery [8]. TCRP is another method for measuring real corneal curvature; however, in addition to studying eyes with GCD2, the study also focused on eyes with other diseases. To the best of our knowledge, no other study has investigated K and TK measured using the IOLMaster in post-PTK cataract surgery.

IOL calculation in post-PTK cataract surgery would be different from IOL calculation in post-PRK or other laser refractive surgeries. The corneal ablation depth in refractive surgery for myopia is larger at the central zone than peripheral zone. On the contrary, PTK has the same ablation depth in the treatment zone, and it would change the anterior curvature of the cornea less than PRK. In the current study, the Barret Ture K formula, known as one of the best accurate methods after refractive surgery, showed larger prediction and absolute errors than other formulas in post-PTK eyes. For this reason, IOL calculation after PTK would need new calculation formula other than those being used after refractive procedures.

This study has certain limitations. In our study, only two eyes had central islands, thus providing insufficient data for statistical analysis. Additional study is required to analyze eyes with central island formation after PTK. Furthermore, the analysis did not include variables such as axial lengths, corneal curvature, and central corneal thickness. In the future, these data should be collected.

Accurate IOL power calculation in post-PTK cataract surgery remains an unresolved challenge. Several rationales are required to avoid refractive error in cataract surgery in eyes with GCD2 that have undergone PTK. Notwithstanding, this outcome suggests that the biometries and formulas using the IOLMaster, especially the Barrett Universal II and SRK/T formulas, are potentially helpful in IOL power calculation without other special methods in patients with GCD2 who have undergone PTK using the flying-spot excimer laser system.

## Figures and Tables

**Figure 1 jcm-12-00584-f001:**
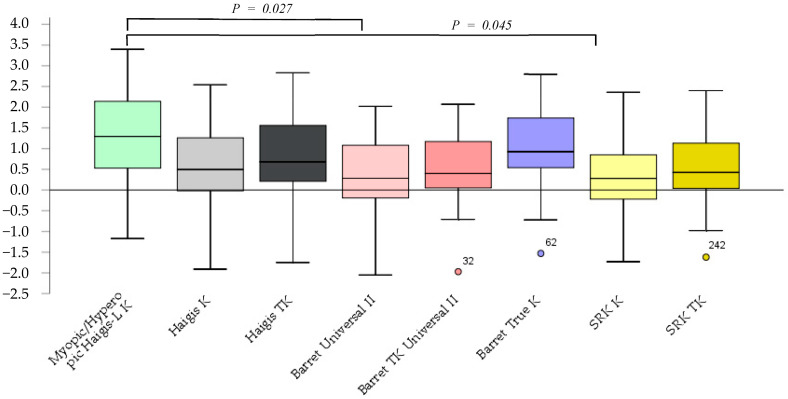
The prediction error after cataract surgery.

**Figure 2 jcm-12-00584-f002:**
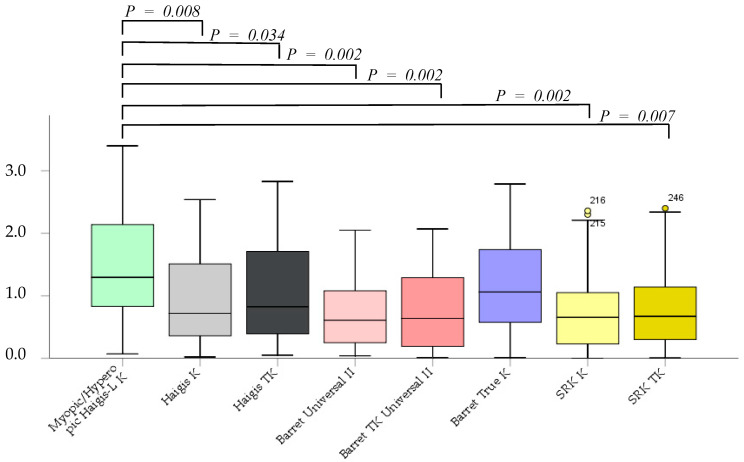
The absolute error after cataract surgery.

**Figure 3 jcm-12-00584-f003:**
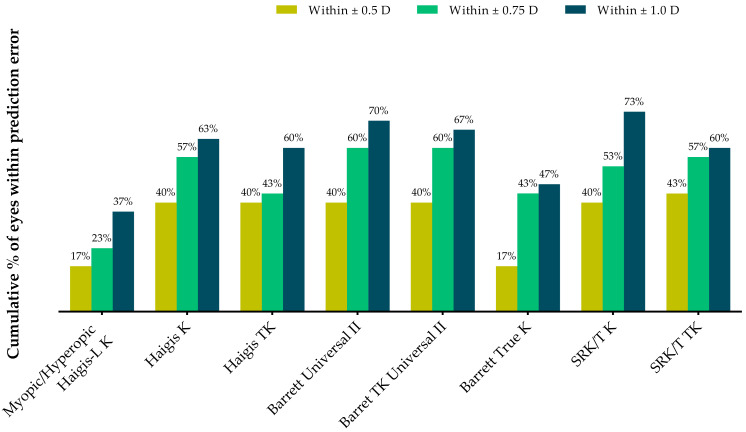
Stacked histogram comparing the proportions of eyes within ±0.25 D, ±0.50 D, and ±1.00 D of the predicted postoperative spherical equivalent refraction at 2 months after surgery.

**Table 1 jcm-12-00584-t001:** Patient characteristics and preoperative biometry.

Parameter	Average ± SD (Range)
Patients/eyes	28/30
Right/left	14/16
Male/female	7/21
Age (years)	68.50 ± 6.84 (57–78)
Mean interval between PTK and cataract surgery (months)	8.37 ± 7.55 (4–33)
Mean IOL power (D)	22.37 ± 3.23 (15–28)
Axial length (mm)	23.89 ± 1.04 (22.47–26.42)
ACD (mm)	3.23 ± 0.35 (2.48–3.84)
Lens thickness	4.43 ± 0.33 (3.69–5.05)
WTW	11.90 ± 0.35 (11.20–12.50)
K1 (D)	42.85 ± 1.57 (39.94–46.53)
K2 (D)	44.24 ± 1.57 (41.57–47.86)
TK1 (D)	42.64 ± 1.66 (39.83–46.31)
TK2 (D)	44.03 ± 1.56 (41.45–47.62)

SD: standard deviation, ACD: anterior chamber depth, WTW: white-to-white, D: diopters, K: standard keratometry, TK: total keratometry.

**Table 2 jcm-12-00584-t002:** The mean prediction and absolute errors and proportions of eyes within ±0.25 D, ±0.50 D, and ±1.00 D of the predicted postoperative spherical equivalent refraction at 2 months after surgery.

	Prediction Error (D)	Absolute Error (D)
Myopic/Hyperopic Haigis-L K	1.31 ± 1.15	1.48 ± 0.91
Haigis K	0.51 ± 1.00	0.88 ± 0.70
Haigis TK	0.74 ± 1.00	0.99 ± 0.75
Barrett Universal II	0.37 ± 0.95	0.78 ± 0.65
Barrett TK Universal II	0.51 ± 0.92	0.80 ± 0.68
Barrett True K	0.97 ± 0.98	1.13 ± 0.78
SRK/T K	0.42 ± 0.98	0.80 ± 0.69
SRK/T TK	0.59 ± 0.96	0.86 ± 0.72

D: diopters, K: standard keratometry, TK: total keratometry.

**Table 3 jcm-12-00584-t003:** The proportions of eyes within ±0.25 D, ±0.50 D, and ±1.00 D of predicted postoperative spherical equivalent refraction at 2 months after surgery.

	Within ±0.5 D (%)	Within ±0.75 D (%)	Within ±1.0 D (%)
Myopic/Hyperopic Haigis-L K	16.7	23.3	36.7
Haigis K	40.0	56.7	63.3
Haigis TK	40.0	43.3	60.0
Barrett Universal II	40.0	60.0	70.0
Barrett TK Universal II	40.0	60.0	66.7
Barrett True K	16.7	43.33	46.7
SRK/T K	40.0	53.3	73.3
SRK/T TK	43.3	56.7	60.0

## Data Availability

All data pertaining to the study was described in the manuscript.

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
