# Peer review of "Comparison of the Predictive Accuracy of Intraocular Lens Power Calculations after Phototherapeutic Keratectomy in Granular Corneal Dystrophy Type 2"

_jcm, 2023, doi:10.3390/jcm12020584_

Round 1
Reviewer 1 Report
Precise prediction of the final refraction in post PTK cataract surgery in addition to the cornea parameter K and PK , the calculation formula or the devices used , it can also suffer from the evaluation of effective lens position(e.g in PEX) or errors in sagittal axis measurement. The study suggests the importance of the Barrett and SRK/T formula when IOL Master biometry is used and thus provides us with very useful conclusions in post PTK cataract surgery.
Author Response
Thank you for your kind comments. We expect this study to contribute as a simple method in post-PTK cataract surgery of GCD 2 patients.
Reviewer 2 Report
1)Looks like a good observational study
2) Do you think there is any difference in the prediction//Absolute error in biometry between post PRK cataract And Post PTK Cataract
3)there are many reports on the subject ... like
a)Intraocular lens power calculation after two different successive corneal
refractive surgeries-American Journal of Ophthalmology Case Reports
Author links open overlay panelZongshengZengXiangyuYeQingzhongChenChangkaiJiaGuangbinZhang
b)IOL power calculations after LASIK or PRK: Barrett True-K biometer-only calculation strategy yields equivalent outcomes as a multiple formula approach
Ferguson, Tanner J. MD; Downes, Rachel A. MD; Randleman, J. Bradley MD
Journal of Cataract & Refractive Surgery: July 2022 - Volume 48 - Issue 7 - p 784-789
doi: 10.1097/j.jcrs.0000000000000883
Author Response
Thanks for the question. Although additional research would find the answer, outcome of IOL calculation might be different from post PTK cataract surgery and post PRK/refractive surgery because ablation nomogram of PTK is different from that of PRK/refractive surgery
Reviewer 3 Report
Good job! The study is very interesting, expecially for surgeons dealing with patients with corneal dystrophies.
----
The paper focuses on the interesting topic of the IOL calculation in patients with previous corneal surgery. the innovative point is to choice a group of patients with GCD2, a rare corneal dystrophy, treated with PTK.
The previous corneal laser treatment in this patients makes the accurately IOL choose very difficult, so comparing multiple formulas can be a great help to the surgeon.
The document is well written and easy to use and it clearly answers the main question.
Author Response
We appreciate your compliment. Although to treat GCD 2 is very difficult, we want to find the simple way to choose IOL in post-PTK cataract surgery.